SCIENCE FORUM

# Author-sourced capture of pathway knowledge in computable form using Biofactoid

**Abstract** Making the knowledge contained in scientific papers machine-readable and formally computable would allow researchers to take full advantage of this information by enabling integration with other knowledge sources to support data analysis and interpretation. Here we describe Biofactoid, a web-based platform that allows scientists to specify networks of interactions between genes, their products, and chemical compounds, and then translates this information into a representation suitable for computational analysis, search and discovery. We also report the results of a pilot study to encourage the wide adoption of Biofactoid by the scientific community.

**JEFFREY V WONG\*, MAX FRANZ, METIN CAN SIPER, DYLAN FONG, FUNDA DURUPINAR, CHRISTIAN DALLAGO, AUGUSTIN LUNA, JOHN GIORGI, IGOR RODCHENKOV, ÖZGÜN BABUR, JOHN A BACHMAN, BENJAMIN M GYORI, EMEK DEMIR\*, GARY D BADER\* AND CHRIS SANDER\***

**\*For correspondence:**
jeffvin.wong@utoronto.ca (JVW);
demire@ohsu.edu (ED);
gary.bader@utoronto.ca (GDB);
sander.research@gmail.com (CS)

**Competing interest:** The authors declare that no competing interests exist.

## Introduction

Biological pathways organize sets of molecular interactions and reactions that underlie cellular processes. This information can be used for experimental design, interpretation of genomics data (*Khatri et al., 2012*), understanding of disease mechanisms (*Chinen et al., 2016*; *Santos et al., 2014*), and identification of therapeutic targets (*Mack et al., 2014*).

To manage, visualize and interpret the large amount of available pathway information, researchers require computational tools such as pathway information resources, structured data representation standards and analysis software (*Demir et al., 2010*; *Franz et al., 2016*; *Jassal et al., 2020*; *Pratt et al., 2015*; *Rodchenkov et al., 2020*; *Shannon et al., 2003*). These computational tools can access the increasing amounts of pathway and interaction data (*Bader et al., 2006*) collected from organizations that either centrally curate the literature (*Gene Ontology Consortium, 2015*; *Jassal et al., 2020*) or crowdsource pathway information from the community of researchers (*Slenter et al.,*

*2018*). Nevertheless, these existing efforts are not able to achieve wide coverage of the rapidly growing scientific corpus (over 1.3 million new PubMed articles per year; *Bornmann and Mutz, 2015*; *Cordero et al., 2016*), arguing for the development of more scalable and sustainable approaches (*Attwood et al., 2015*; *Imker, 2018*).

For reasons of efficiency and accuracy, structured knowledge of biomedical discoveries could be provided directly by the original authors of research reports, rather than post-hoc, through the curation efforts of knowledge base teams. By analogy, molecular 3D structures submitted directly by authors to the Protein DataBank (PDB) have become a key resource among structural biologists (*Berman et al., 2000*), and similar community practice led to direct submission of DNA sequences and transcriptomics information to public databases. Indeed, the importance and value of having such research outputs available in a community resource are underscored by the fact that data deposition is often a requirement of publication and funding. In contrast, there are few efforts and little technology to support direct

submission by authors of biological pathway information and related knowledge in computable form.

Here we introduce Biofactoid ([biofactoid.org](biofactoid.org)), a web-based software system that empowers authors to capture and share structured human- and machine-readable summaries of molecular-level interactions that are supported by evidence in their publications. Without such a curation support tool, the onus would be on authors to handle a series of complex tasks involved in converting their knowledge to a computational form and depositing it in a suitable knowledge base. To overcome this and other significant barriers to computable knowledge acquisition and sharing, we developed Biofactoid to ease pathway curation and to rapidly generate expressive, structured representations with minimal user training. Structured knowledge newly acquired in this way becomes part of the global pool of pathway knowledge and can be shared in resources such as Pathway Commons (*Rodchenkov et al., 2020*), Network Data Exchange (NDEx) (*Pratt et al., 2015*), and STRING (*Szklarczyk et al., 2017*), to enhance information discovery and analysis.

Authors can use Biofactoid to share structured information from research articles both as part of the publication process and outside of it. Personal email invitations sent to authors as part of a pilot outreach effort resulted in them contributing over 95 articles to Biofactoid from a range of journals; collectively these pathways have described a total of 426 entities (340 unique) from eight organisms (*H. sapiens, M. musculus, R. norvegicus, D. rerio, S. cerevisiae, D. melanogaster, C. elegans, E. coli*) involved in 287 interactions. The development of Biofactoid and related computational tools helps support human-computer communication and inference algorithms in an ecosystem in which scientific reasoning is increasingly assisted by broad and deep knowledge computation.

## Results

### *Data sharing workflow*

Biofactoid enables molecular-level detail of biological processes reported in articles to be shared in a structured format accessible to humans and computers. Interactions (binding, post-translational modification and transcription/translation) involving molecules of various types (proteins, nucleic acids, genes, or chemicals such as metabolites and drug compounds)

can be represented. Users begin by entering the article title, or identifier (e.g. PubMed identifier or Digital Object Identifier). Next, article metadata, including authors, abstract and journal issue, are automatically retrieved. Next, users draw a network of biological entities and interactions using the Biofactoid curation tool, which has an easy-to-use interface similar to graphical illustration software (e.g. Microsoft Powerpoint, Adobe Illustrator) which will be familiar to users, but with the added ability to generate structured data from the author-drawn biological pathway. The major features of the Biofactoid curation tool include support for:

1. Molecular entities. Genes, gene products and chemicals are created in the network using an "Add a gene or chemical" tool, which creates a node (circle) that users label (*Figure 1A*). Biofactoid automatically matches the label with a record in an external database: NCBI Gene for genes and their products and ChEBI for small molecules (*Brown et al., 2015*; *Hastings et al., 2016*). This match represents the top hit of a search based on the similarity between a user's label and a database record's list of common names and synonyms; for genes, organisms are given priority based upon the organism of genes previously added to the network. Users can update the match by selecting another organism (e.g. human or mouse p53) or update the automatically inferred gene product type (i.e. RNA or protein). Alternatively, users may assign an alternate database entry from a list of search results. The system supports human and selected model organisms (*M. musculus, R. norvegicus, S. cerevisiae, D. melanogaster, E. coli, C. elegans, D. rerio, and A. thaliana*). The system is fast (response times of 100 milliseconds or less) and can be easily extended, for instance, to support SARS-CoV and SARS-CoV-2, which we added in response to the burst of publications related to the COVID-19 global pandemic (*Ostaszewski et al., 2020*). The matching process is also accurate, as measured using tests that use entity names from research articles as queries and assessing the quality of the search result. In over 90% of the cases, the correct result was first among search hits and was among the top 10 in over 97% of cases. Thus, Biofactoid can offer an accurate database identifier match using only the author-provided entity name. This represents a major advance in usability compared to traditional gene and chemical name querying systems that only work with exact name string matching and

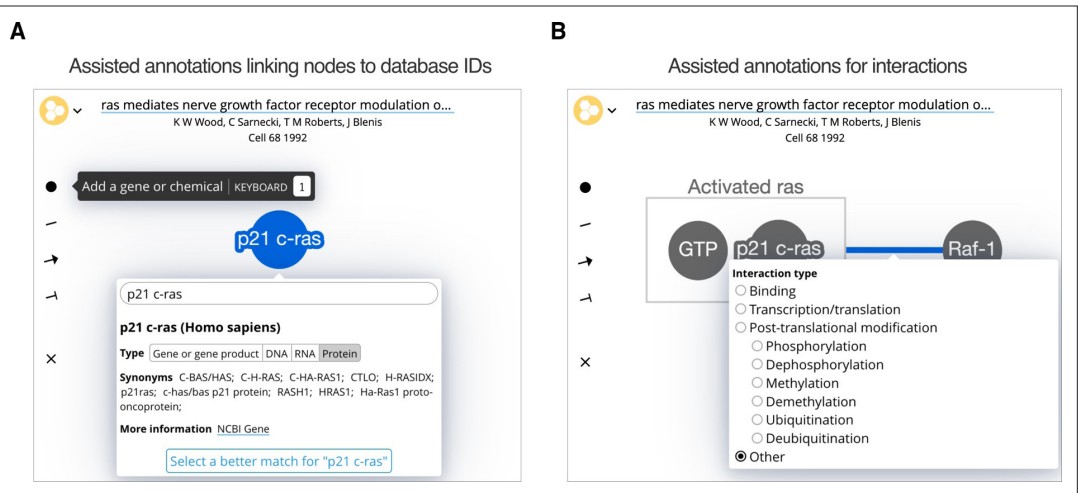

**Figure 1.** The Biofactoid curation tool. Curation in Biofactoid involves drawing a network of relationships between genes or chemicals. (**A**) Genes and chemicals are represented by circles (nodes, highlighted in blue) where users provide a label, the type of gene product, and the organism. A search engine matches the label to a corresponding record from a database of genes or chemicals. (**B**) Relationships are represented by connecting genes, chemicals and/or rectangular complexes (shown in grey) with plain lines (when neither activation nor repression occur), arrows (to indicate activation), or 'T-bars' (to represent repression). Users select the mechanism that best describes the interaction. Complexes are represented as genes and/or chemicals enclosed by a box (e.g. the grey box labelled "Activated ras").

are often slow or unreliable. In fact, we were forced to build our own system after trying to use major existing systems that turned out to have high failure rates for our use case.

2. Interactions. A "Draw interaction" tool lets users link two nodes by clicking one and dragging to the other (*Figure 1B*). Edges can have type and direction: a pointed arrow indicates stimulation or activation; a 'T-bar' arrowhead indicates inhibition or repression; and an undirected line indicates any other interaction. The interaction mechanism can be refined by selecting from a list that includes binding, transcription/translation, or common forms of post-translational modifications.

3. Complexes. Molecular complexes can be created by dragging genes or chemicals close to each other, resulting in a box that encloses them, which can also be labeled (*Figure 1B*).

4. Context. A context input box that encourages authors to "List terms that describe the context (e.g. T cell, cancer, genome stability)". Allowing authors to provide a list in free text form has the benefit of accommodating any terms or concepts they consider important for specifying the research context.

5. Automatic saving and co-editing. Pathway representations in Biofactoid are automatically saved as changes are made and

a live-sync capability enables multiple authors to collaboratively edit the same pathway, analogous to Google Docs.

Once complete, the pathway data is validated by pressing a "Submit" button. At this point, the user may address any potential quality issues flagged by the system (e.g. unlabelled nodes, empty document) before confirming their submission. Submitted pathways are shared publicly with the research community and users retain the ability to edit their pathway via the link they received at the start of the workflow. Authors who wish to have their submissions removed from Biofactoid can do so by contacting the support team (support@biofactoid.org).

### Data sharing and exploration

When research findings are shared through Biofactoid in a structured and computable manner, curated knowledge is connected to, and becomes part of, a collective pool of computable knowledge that the community can access in different ways. Visitors to the Biofactoid website (biofactoid.org) may browse recently added articles, and a graphical abstract of each new submission is posted to Twitter (twitter.com/biofactoid). Each entry is automatically linked to its associated authors, article information and structured data and presented in an interactive Biofactoid Explorer web app (*Figure 2A*).

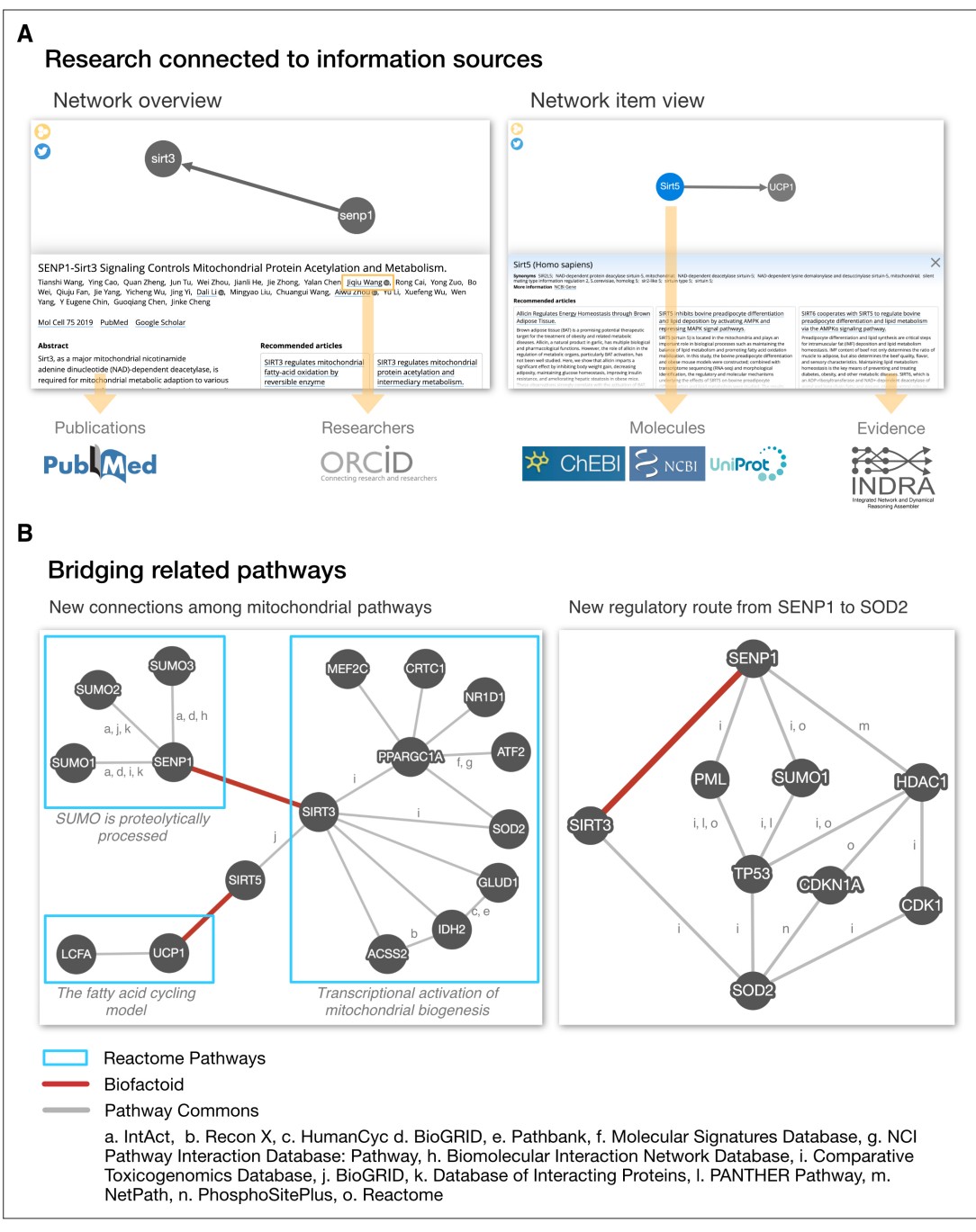

**Figure 2.** Biofactoid data is connected to information sources and establishes a bridge between related pathways. (**A**) The Biofactoid Explorer is an interactive web app that publicly presents each author-curated entry alongside their article. Yellow arrows indicate how curated information is connected to outside knowledge bases. A "Network overview" (left) displays information about the article and pathway as a whole; a "Network item view" (right) displays information for a selected item (e.g. interaction, protein). (**B**) Biofactoid data establishes a bridge between related pathways described by structured biological knowledge from distinct data sources. Two author-curated interactions submitted to Biofactoid (red edges) bridge previously distinct pathways from the Reactome Pathway Database involved in mitochondrial biogenesis (left) and provide a new, more direct regulatory route between two mitochondrial genes (right). Pathway and interaction information was provided by Pathway Commons (pathwaycommons.org), a web resource that provides a single point of access for multiple public interaction and pathway databases. Details regarding the generation of these networks can be found in the section "Visualization of network data across sources" in Materials and methods.

Users can select any entity in the pathway diagram to see more information about genes (via NCBI Gene database links) and chemicals (via ChEBI database links). To support attribution, each author of an article is automatically linked to their profile in the Open Researcher and Contributor ID database (ORCID; orcid.org). Finally, a set of "Recommended articles" are generated to help the user explore similar knowledge in the literature.

To generate these recommendations, we first retrieve an article's references, articles it is cited by and "Similar articles" from PubMed using the NCBI Entrez Programming Utilities service (*Wheeler et al., 2006*). These are combined with articles that mention or provide evidence for interactions involving any of the molecular entities in the Biofactoid entry, which are accessed from curated pathway and interaction databases as well as resources that aggregate interaction information directly from the biomedical literature using natural language processing (NLP) tools (e.g. REACH, INDRA) (*Gyori et al., 2017*; *Valenzuela-Escárcega et al., 2018*). Recommended articles are ranked by determining how similar their titles and abstracts are to that of the Biofactoid article, using a deep-learning-based method we developed (*Giorgi et al., 2021*). The list of recommendations is context-sensitive in that articles are shown relevant to the part of the pathway selected by a user.

Beyond the "Explorer", Biofactoid data represented in the formal BioPAX ontology is interoperable with external pathway and interaction knowledge using technologies for BioPAX processing and analysis, including Paxtools (*Demir et al., 2013*) and cPath2 (*Cerami et al., 2006*). This permits author-contributed information to be connected to existing information, and leverages tools for searching, visualizing and analyzing data across different sources.

For example, researchers can ask the question: "Which known pathways are related to an interaction in Biofactoid?" As an example answer to this question, a search of Pathway Commons shows how two interactions made computable by authors using Biofactoid (i.e. "*SENP1 activates SIRT3*", *Wang et al., 2019a*; "*SIRT5 activates UCP1*", *Wang et al., 2019b*) unify previously disconnected pathways for mitochondrial biogenesis from the Reactome Pathway Database (*Jassal et al., 2020*; *Figure 2B*; see the section "Visualization of network data across sources" in Materials and methods). Researchers can also ask: "How are two genes related?" As another query example, a search of Pathway

Commons shows how an interaction contributed by an author using Biofactoid ("*SENP1 activates SIRT3*", *Wang et al., 2019b*) provides a new, more direct regulatory pathway linking the mitochondrial proteins SENP1 and SOD2. Thus, Biofactoid directly contributes to building a more complete and comprehensive collection of biological pathways.

### User testing and pilot study involving authors and journals

We tested the Biofactoid software over multiple iterations with researchers from the authors' institutions, journal editors and finally pathway authors, updating the software each time based on user feedback. This process supported two major goals: to improve the user experience of the software for authors, who are typically unfamiliar with curation and structured data concepts, and to develop a model for integrating Biofactoid into the publication process with journals. We performed user testing, with a user-centred design process, of a prototype of the Biofactoid curation tool (*Norman, 2001*; *Norman and Draper, 1986*). We improved our software based on this feedback and then repeated this process a second time (see the section "User testing and pilot study" in Materials and methods). We also gathered more informal feedback that was used to improve the system.

Once Biofactoid software achieved a sufficient level of sophistication and completeness, satisfying many of these initial users, we engaged journal editors to determine the feasibility of using Biofactoid to capture information from authors. Our pilot study consisted of three phases (*Figure 3*). Phase I introduced Biofactoid to journal editors via an "author-simulation", which began with a mock email invitation asking the editor to use Biofactoid to curate a selected article, and ending when a pathway from that article was input into the system by the editor using the Biofactoid web-based user interface. Phase II involved sending email invitations to a small number (N = 15) of authors from a single journal, with multiple successful responses, proving that authors can use the system without any direct support from the Biofactoid team or journal. Phase III measured engagement rate by emailing 260 authors of articles with content suitable for Biofactoid from 16 selected journals (*Table 1*).

Roughly 8.5% of these authors successfully shared their research in Biofactoid, proving that many authors can and will use Biofactoid

**Table 1.** Prevalence of articles with pathway knowledge suitable for Biofactoid.

| ISSN | Journal* | Coverage[2] [Vol. (Issue)] | Articles screened | Hits | % Hits |
|------|----------|----------------------------|-------------------|------|--------|
| 2211–1247 | Cell Reports | 30(1) - 32(11) | 953 | 109 | 10.3 |
| 1097–4164 | Molecular Cell | 73(1) - 79(6) | 725 | 85 | 10.5 |
| 1549–5477 | Genes & Development | 34(1-2) - 34(17-18) | 93 | 15 | 13.9 |
| 1476–4679 | Nature Cell Biology | 22(4) - 22(9) | 84 | 10 | 10.6 |
| 1083–351 X | Journal of Biological Chemistry | 295(31) - 295(37) | 210 | 21 | 9.1 |
| Total | - | - | 2065 | 240 | - |
| Weighted Average | - | - | - | - | 10.4 |

*Only journals in which at least 80 'hits' were identified were included. A 'hit' is an article that provides direct evidence for a molecular interaction that can be captured by Biofactoid. The ability of Biofactoid to capture an interaction depends upon the type of bioentities, the relationship types and organisms described in the article. In total, articles from 16 journals were screened: EMBO; Molecular and Cellular Biology; Cell; Cancer Cell; iScience; J Biol Chem; Cell Metabolism; Science; Nature Genetics; Science Signaling; Science Advances; Immunity; Cell Reports; Molecular Cell; Genes & Development; Nature Cell Biology. [2]Coverage indicates the span of journal issues that were included. Only primary research articles from each issue were screened.

**Phase I**

Purpose: Introduce journal editors to Biofactoid

Method: Software walk-though with editors

Outcome: Editor go-ahead for Phase II

**Phase II**

Purpose: Proof of concept for workflow and software with authors

Method: Manually screen articles from one journal and send email invitations to authors (N=15)

Outcome: Authors could submit successfully (3 submissions)

**Phase III**

Purpose: Determine author engagement rate

Method:  Manually screen articles from 16 journals and send email invitations to authors (N=260)

Outcome: 8.5% engagement (22 submissions)

**Figure 3.** Biofactoid pilot study. A three-phase pilot tested the feasibility of Biofactoid and involved journal editors and authors of research articles. Phase I and II involved editors and authors whose articles were recently published. In Phase III, 2,065 published articles were screened and authors of suitable articles were invited to Biofactoid. The articles screened were from 16 journals (Table 1).

(*Figure 2A*). We also found that 10% of articles across selected journals are suitable for inclusion in Biofactoid based on the current set of supported concepts (*Table 1*). In addition to proving that Biofactoid can be independently used by authors to capture pathway knowledge, these results indicate a willingness among authors in the research community to use the system. These also demonstrate that the information captured by authors may not be present in any existing pathway database and helps connect previously unconnected entities and pathways in these databases (*Figure 2B*). We also observed that pilot study authors invariably chose to represent only their novel findings, rather than previously known pathway information found in their background sections.

## Discussion

Biofactoid focuses on the capture of published pathway information in computational form to augment discovery, attribution and communication of scientific knowledge. In developing our strategy, we have considered how technology can best be used to aid authors. The result is a generic approach that can be extended to capture other types of biological knowledge, at the source of knowledge creation, in computable form. To be successful, Biofactoid development has focused on providing key elements of efficiency, incentives and better technology for human-computer interaction.

The computable knowledge capture model proposed here includes formal knowledge representation using ontologies, easy to use curation support software that links molecules to corresponding database identifiers (normalization), submission to widely available knowledge resources, a connection with the publication process and author attribution. This model expands on prior work defining digital

abstracts and building crowdsourcing efforts for pathway data and software (*Ceol et al., 2008*; *Gerstein et al., 2007*; *Leitner et al., 2010*; *Liechti et al., 2017*; *Slenter et al., 2018*; *Table 2*). In principle, Biofactoid technology can be extended to support any network of entities or concepts and their relationships (i.e. a knowledge graph) collaboratively built by a community of knowledge generators.

A key assumption underlying Biofactoid is that scalable knowledge capture can be achieved by community curation. However, crowd-sourcing of biological research findings presents unique challenges that must be considered if widespread use is to be realised. First, the biological research literature contains a wide array of biological entities and concepts, with descriptions that range from detailed to abstract. As support for more biological entity types are added, a curation system can become more complex, increasing the barrier to entry and making it more likely users will require training to use it effectively. Thus, a successful strategy should support wide coverage of biological concepts while minimising complexity.

In a related issue, biological researchers are accustomed to communicating pathway concepts using visual depictions that employ symbols and conventions, which, although comprehensible to the community, are often ad hoc, abstract and potentially ambiguous (e.g. "A activates B"). Thus, a successful curation system should accommodate ways that biologists articulate concepts, while also retaining the ability to represent the information in a rigorous and computable manner.

A second major challenge to achieving uptake is raising awareness and communicating the goals of the project within the research community. Pathway information is qualitatively distinct from items that researchers are accustomed to

**Table 2.** Comparison of non-centrally curated biocuration projects.
Comparison of projects that support community curation of pathway and interaction knowledge as their primary concern.

| Project | Scope | Source | Integrated curation tool | Automatic entity recognition | Single-article oriented | Ref. |
|---|---|---|---|---|---|---|
| Biofactoid | Pathway | Author | ✓ | ✓ | ✓ | This study. |
| Structured Digital Abstract (FEBS Letters) | Protein-Protein | Author | - | - | ✓ | *Ceol et al., 2008*; *Leitner et al., 2010*; *Gerstein et al., 2007* |
| WikiPathways | Pathway | Anyone | - | - | - | *Slenter et al., 2018* |
| SourceData | Figure | Author | ✓ | - | ✓ | *Liechti et al., 2017* |

depositing (e.g. DNA sequencing data), as pathways are models that represent the outcome or interpretation of multiple experiments. Ideally, potential contributors should be able to independently determine whether their articles are suitable for Biofactoid.

A third challenge to realising wide uptake is incentivizing authors to contribute the knowledge in their article. In particular, authors will be more likely to contribute if the information they enter will be useful, either directly to themselves and/or others in their community. Scalable and sustainable growth is more likely to be achieved when researchers directly use and benefit from Biofactoid information and then decide to contribute themselves.

We designed Biofactoid to enable researchers to express details of their research quickly and effectively, with minimal instruction. To preserve ease of use, we have purposely limited both the scope of biological concepts and the granularity of detail that can be captured, though this can be expanded in the future. With regard to entities, only major model organisms and SARS-CoV(–2) are supported; we have not included more abstract concepts like cellular processes (e.g. apoptosis) or fine details like reaction stoichiometry, molecular sub-regions (e.g. promoter region) and modification (i.e. amino acid position, mutations). With regard to interactions, the system does not explicitly support biological concepts that involve spatial (e.g. cellular location) or temporal considerations (e.g. ordered recruitment of proteins).

The aforementioned details are supported by the underlying BioPAX ontology and additional concepts (e.g. global biological context, that is, the environment in which pathway components act) can be supported by referencing other controlled vocabularies or databases from the BioPAX model. While the current set of features constrains the scope of capturable research facts, it enables Biofactoid to be accessible by a wide audience. Nevertheless, future work will expand the range of biological concepts supported.

Distributing curation effort among authors has the potential to be efficient but also accurate because of their first-hand knowledge of the research. To ensure authors can use Biofactoid effectively we developed an automatic entity recognition tool that accurately matches the names of genes and chemicals provided by authors to an entity database record. We also flag potential issues with author input at submission so they can be corrected immediately (e.g. unlabelled entities, genes from multiple organisms).

Finally, we evaluated our system through user testing to improve the speed and accuracy with which users can curate interactions described in text. In the future, we will develop strategies to monitor the accuracy of pathway data post hoc *and* identify and amend inaccuracies we discover.

To monitor data accuracy, we will seek out experts to independently rate pathway items, which is an approach that has been successfully applied to problems in computer-aided diagnostics and healthcare (*McHugh, 2012*). Specifically, we will recruit at least two experienced curators of pathway knowledge to assign quantitative ratings for a sample of Biofactoid pathways with respect to accuracy of database identifiers for entities, accuracy of interactions and completeness (i.e. missing/extraneous elements). Curator scores will be compared to estimate variability. We will also enable users to take action when they identify inaccurate information by contacting the article author to ask them to edit and correct the record. Failing this, we will enable users to directly report an issue with any particular pathway to the Biofactoid support team and will reserve the right to lock editing of pathway items, mark them as disputed or remove them from the collection.

A convenient time to capture knowledge is during publication, when authors - the primary source of knowledge - are most aware of the details of their report and when they are typically asked to deposit other types of data (e.g. DNA sequencing or gene expression data). As with any innovative approach, introducing the system to editors, journals and their respective authors requires an investment of effort for all parties. It will likely be necessary to directly engage individual journals, to familiarize them with the system, and accommodate new requirements they may have. There are many ways to offer Biofactoid to authors, including, for instance: integrating it with the peer-review process, requiring it as a condition of publication or recommending it to authors post-publication-acceptance.

As a demonstration, our pilot study engaged publishers to establish requirements for using Biofactoid within their publication pipeline, ideally as a condition for acceptance, and to drive greater awareness of the system among the wider readership. The pilot study also revealed that directly inviting authors of suitable articles to contribute, even after publication, is a viable engagement strategy. Ideally, we would invite all authors who publish suitable content to use Biofactoid, however, the rapid pace of publication makes it impractical for the Biofactoid team to manually screen articles, arguing for the

development of systems (e.g. using NLP) to automate article triage. If successful, this approach could be used to regularly identify relevant articles from the pool of all new entries indexed by PubMed.

Another approach to reach users is to notify those whose research articles are referenced by information curated in Biofactoid. For this purpose, we have developed a system that automatically notifies authors when their research is linked to papers curated in Biofactoid (e.g. by citations or because of related content) so that they may further explore this information and curate their own pathway knowledge. More generally, we intend to engage the research community by cultivating partnerships with knowledge database organizations including those that curate information about articles (PubMed), biomolecules (e.g. UniProt, NCBI, ChEBI), pathways and interactions (e.g. Reactome, STRING), model organisms (e.g. *Saccharomyces* Genome Database [*Lang et al., 2018*]) and researchers (e.g. ORCID).

To improve the efficiency and utility of Biofactoid, we are developing NLP technology to support authors in using Biofactoid, as well as to enable the representation of pathways in textual form (*Giorgi et al., 2019*; *Giorgi and Bader, 2018*; *Giorgi and Bader, 2020*; *Valenzuela-Escárcega et al., 2018*). To better accommodate the way individual users prefer to communicate, the system will accept both graphical and textual entry of pathway information as well as automated conversion between these two forms. Assistants will support users to rapidly compose their network, enabling them to add new information from a list of interactions identified in their article by NLP technology (*Gyori et al., 2017*). Further development of NLP methods for the reliable extraction of pathway information from the publication full-text, combined with development of new tools for curation and quality control, will help realize broad and accurate coverage of pathway knowledge in computable form.

We are also improving the search and exploration functions of the system, ensuring Biofactoid information is connected to other useful knowledge, and that related information is easily accessible starting from a Biofactoid entry. In the future, we envision that information entered by an author in Biofactoid will serve as a custom query that can be used to regularly notify the author of new information (e.g. from other publications) related to their interests, such as interacting molecules and phenotypes. This work provides a basis for the development of new technologies to make scientific knowledge more computable and accessible and help researchers identify information within the rapidly growing scientific corpus.

## Materials and methods

### *Software implementation*
Biofactoid is written in JavaScript. The backend server uses a microservice architecture, with Node.js, Express and RethinkDB. Client-server data synchronization, supporting automatic saving and concurrent editing, uses websockets and a model similar to differential synchronization (*Fraser, 2009*). The front end uses React and Cytoscape.js (*Franz et al., 2016*), for network drawing and is optimized for desktop and mobile devices. An administrative dashboard, as well as user curation workflow automation features (e.g. automatic email generation) are integrated into the Biofactoid system to aid system scalability.

### *Visualization of network data across sources*
The network visualizations which summarize a search of known pathways and interactions related to interactions captured by Biofactoid (*Figure 2B*) were created by combining data available in Biofactoid and Pathway Commons (version 12; *Rodchenkov et al., 2020*). Data for the network with subheading "New connections among mitochondrial pathways" was obtained by searching pathwaycommons.org for the genes SENP1, SIRT3, UCP1 and SIRT5, which are part of the Biofactoid entries displayed in *Figure 2A*. In the search results, the Reactome pathways entitled "SUMO is proteolytically processed", "Transcriptional activation of mitochondrial biogenesis" and "The fatty acid cycling model" were each downloaded as a Simple Interaction Format (SIF) file. Also in the search results, the "Interactions" panel was selected and the network was downloaded as a SIF file. Using Cytoscape ( cytoscape.org) (*Shannon et al., 2003*), each SIF file was imported as a new network and the four individual networks were merged. The Biofactoid interactions from *Figure 2A* were added to this merged network which was manually laid out and styled to achieve the final result. Data underlying the network with figure subheading "New regulatory route from SENP1 to SOD2" was obtained by querying the Pathway Commons SIF Graph web service (http://www.pathwaycommons.org/sifgraph/swagger-ui.html) to identify regulatory

paths between SENP1 and SOD2, in which the maximum number of interactions separating the two genes (i.e. limit) is 3. This resulting SIF file was imported into Cytoscape and then manually laid out and styled to achieve the final result.

### User testing

Two rounds of user-centered testing were performed, once following the development of the initial prototype of the Biofactoid curation tool, and once again after further development of the tool. Test participants were drawn from volunteers who responded to individual email invitations. In each round of testing, participants were provided with a one-page handout that provided an overview of the project, brief instructions on how to use the curation tool, followed by one or more text excerpts describing a biological interaction that they were asked to input. Ten researchers participated, including faculty members (n = 2), graduate students (n = 3) and research staff (n = 5). Participants used the software on an individual, in-person basis from a provided computer, during which screen and voice were recorded. All participants provided written consent to volunteer and have their session recorded. We also elicited user feedback both in-person and by email for a version of Biofactoid able to support the entire "Data sharing workflow", including: user onboarding material presented on the biofactoid.org homepage; a system to identify articles based on information entered by authors (i.e. title); the curation tool; and the Biofactoid Explorer. Those providing feedback for the "Data sharing workflow" included three curators from biological databases (Reactome and UniProt); participants were asked to input an article of their choosing, starting at the homepage.

### Pilot study

A group of two editors from Molecular Cell and Cell Systems volunteered to evaluate Biofactoid in response to a direct invitation. A pilot study was designed with three phases. Phase I introduced the software to the journal editors to determine whether it met basic requirements for capturing pathway knowledge in articles in their journals and identify any issues precluding interaction with authors. To do this, we simulated the author's experience using the Biofactoid curation tool to input pathway information. First, an editor and a member of the Biofactoid team selected a recently published article from the Molecular Cell, "Online Now" (https://www.

cell.com/molecular-cell/newarticles) website, based on a single criterion: the article reported evidence directly supporting at least one interaction type supported by Biofactoid. Next, editors were sent a mock email invitation addressed to the selected article's corresponding author. Invitations contained brief instructions on system use and a link to the Biofactoid curation tool, which had the article metadata already filled for the corresponding Biofactoid document. Finally, an editor was observed using the curation tool to enter the pathway information contained in the article remotely via video-conferencing software. A feedback session was carried out with two editors immediately following the simulation and a subsequent video-conference was scheduled to capture additional recommendations for the software and author workflow.

Phase II validated the Biofactoid software and the workflow with authors. First, a set of suitable articles that were recently published by Molecular Cell (Online Now) were selected with journal editors. Second, email invitations addressed to the corresponding author of each article were sent from the institutional email account of a Biofactoid team member. As in Phase I, emails contained a link to the curation tool with their article information already filled in. To our knowledge, these authors had no prior contact with the journal with regard to Biofactoid. Follow-up reminders were sent after one week to authors who had not used the system. We repeated this process with two additional groups of articles (ten total) from Molecular Cell (Online Now) that were selected by the Biofactoid team.

Phase III examined the author participation rate following an invitation. First, we selected a total of sixteen journals, with a preference towards those with a mandate for molecular biology and mechanistic studies backed by biochemical-level evidence. Journals with broad and diverse readership (e.g. Nature, Science; *Table 1*) were also included. Next, articles from entire journal issues were screened by a Biofactoid team member (JVW) for studies reporting evidence of one or more biological interactions compatible with Biofactoid. As before, email invitations were addressed to an author of the article. In contrast to previous emails, the recipient was only provided with a link to the biofactoid.org homepage in order to begin the data sharing workflow (no pre-fetching of manuscript metadata). A maximum of two weekly reminder emails were sent to those that had not submitted their pathway information. We manually analysed user feedback to identify bugs and aspects of

the system that were not intuitive or prevented users from entering data and prioritized those to be addressed with Biofactoid software improvements.

### Software availability
Biofactoid is available to biomedical researchers for data sharing and exploration at biofactoid. org. To support bioinformaticians and software developers, all user-contributed pathway data is openly accessible in multiple standard formats: JavaScript Object Notation (JSON) for raw data; Systems Biology Graphical Notation Markup Language (SBGN-ML) pathway visualization format using the Process Description notation (*Le Novère et al., 2009*; *van Iersel et al., 2012*); and BioPAX (*Demir et al., 2010*). All code, documentation and data are open source and freely available through GitHub (github.com/PathwayCommons/factoid); containerized software components are freely available on DockerHub (hub.docker.com/r/pathwaycommons/factoid) enabling others to build on and improve the Biofactoid software.

### Acknowledgements

We thank Quincey Justman, Miao-Chih Tsai, and Anita DeWaard for feedback on making Biofactoid useful for editors and authors; the Reactome database team for support and feedback on the curation workflow; Alfonso Valencia, and Miguel Vazquez for early support; and the many community members in Toronto, Boston, Portland, and beyond for feedback on the design and concept of Biofactoid.

**Jeffrey V Wong** is in The Donnelly Centre, University of Toronto, Toronto, Canada
jeffvin.wong@utoronto.ca
http://orcid.org/0000-0002-8912-5699

**Max Franz**  is in The Donnelly Centre, University of Toronto, Toronto, Canada
http://orcid.org/0000-0003-0169-0480

**Metin Can Siper** is in the Computational Biology Program, Oregon Health and Science University, Portland, United States
http://orcid.org/0000-0002-7556-093X

**Dylan Fong** is in The Donnelly Centre, University of Toronto, Toronto, Canada

**Funda Durupinar** is in the Computer Science Department, University of Massachusetts Boston, Boston, United States
http://orcid.org/0000-0002-4915-6642

**Christian Dallago** is in the Department of Cell Biology and the Department of Systems Biology, Harvard Medical School, Boston, United States and in the Department of Informatics, Technische Universität München, Garching, Germany
http://orcid.org/0000-0003-4650-6181

**Augustin Luna** is in the Department of Cell Biology, Harvard Medical School; in the Department of Data Sciences, Dana-Farber Cancer Institute; and in the Broad Institute, Massachusetts Institute of Technology and Harvard University, Boston, United States
http://orcid.org/0000-0001-5709-371X

**John Giorgi** is in The Donnelly Centre, University of Toronto, Toronto, Canada
http://orcid.org/0000-0001-9621-5046

**Igor Rodchenkov** is in The Donnelly Centre, University of Toronto, Toronto, Canada

**Özgün Babur** is in the Computer Science Department, University of Massachusetts Boston, Boston, United States
http://orcid.org/0000-0002-0239-5259

**John A Bachman** is in the Laboratory of Systems Pharmacology, Harvard Medical School, Boston, United States
http://orcid.org/0000-0001-6095-2466

**Benjamin M Gyori** is in the Laboratory of Systems Pharmacology, Harvard Medical School, Boston, United States
http://orcid.org/0000-0001-9439-5346

**Emek Demir** is in the Computational Biology Program, Oregon Health and Science University, Portland, United States
demire@ohsu.edu
http://orcid.org/0000-0002-3663-7113

**Gary D Bader** is in The Donnelly Centre, The Department of Computer Science and the Department of Molecular Genetics, University of Toronto; The Lunenfeld-Tanenbaum Research Institute, Mount Sinai Hospital; and the Princess Margaret Cancer Centre, University Health Network, Toronto, Canada
gary.bader@utoronto.ca
http://orcid.org/0000-0003-0185-8861

**Chris Sander** is in the Department of Cell Biology, Harvard Medical School; in the Department of Data Sciences, Dana-Farber Cancer Institute; and in the Broad Institute, Massachusetts Institute of Technology, Harvard University, Boston, United States
sander.research@gmail.com
http://orcid.org/0000-0001-6059-6270

*Author contributions:* Jeffrey V Wong, Conceptualization, Data curation, Formal analysis, Investigation, Methodology, Project administration, Software, Validation, Visualization, Writing – original draft, Writing – review and editing; Max Franz, Conceptualization, Data curation, Formal analysis, Investigation, Methodology, Project administration, Software, Writing – review and editing; Metin Can Siper, Data curation, Software; Dylan Fong, Software; Funda Durupinar, Software; Christian Dallago, Conceptualization, Methodology, Project administration, Writing – review and editing; Augustin Luna, Conceptualization, Investigation, Methodology, Writing – review and editing; John Giorgi, Software;

Igor Rodchenkov, Software; Özgün Babur, Conceptualization, Methodology, Supervision; John A Bachman, Resources; Benjamin M Gyori, Resources; Emek Demir, Conceptualization, Funding acquisition, Methodology, Project administration, Supervision, Writing – review and editing; Gary D Bader, Conceptualization, Funding acquisition, Methodology, Project administration, Supervision, Writing – review and editing; Chris Sander, Conceptualization, Funding acquisition, Methodology, Project administration, Supervision, Writing – review and editing

**Competing interests:** The authors declare that no competing interests exist.

**Ethics:** Human subjects: Participants of user testing provided written consent to volunteer, have their testing sessions recorded and have quotes obtained in the session published.

## Funding

| Funder | Grant reference number | Author |
| --- | --- | --- |
| National Human Genome Research Institute | U41 HG006623 | Jeffrey V Wong<br>Max Franz<br>Metin Can Siper<br>Dylan Fong<br>Funda Durupinar<br>Christian Dallago<br>Augustin Luna<br>John M Giorgi<br>Igor Rodchenkov<br>Özgün Babur<br>Emek Demir<br>Gary D Bader<br>Chris Sander |
| National Human Genome Research Institute | U41 HG003751 | Jeffrey V Wong<br>Max Franz<br>Metin Can Siper<br>Dylan Fong<br>Funda Durupinar<br>Christian Dallago<br>Augustin Luna<br>John M Giorgi<br>Igor Rodchenkov<br>Özgün Babur<br>Emek Demir<br>Gary D Bader<br>Chris Sander |
| National Human Genome Research Institute | R01 HG009979 | Max Franz<br>Gary D Bader |
| National Institute of General Medical Sciences | P41 GM103504 | Jeffrey V Wong<br>Max Franz<br>Metin Can Siper<br>Dylan Fong<br>Funda Durupinar<br>Christian Dallago<br>Augustin Luna<br>John M Giorgi<br>Igor Rodchenkov<br>Özgün Babur<br>Emek Demir<br>Gary D Bader<br>Chris Sander |
| Defense Advanced Research Projects Agency | Big Mechanism | Metin Can Siper<br>Funda Durupinar<br>Özgün Babur<br>John A Bachman<br>Benjamin Gyori<br>Emek Demir |
| Defense Advanced Research Projects Agency | Communicating with Computers | Metin Can Siper<br>Funda Durupinar<br>Özgün Babur<br>John A Bachman<br>Benjamin Gyori<br>Emek Demir |
| Defense Advanced Research Projects Agency | ARO W911NF-14-C-0119 | Metin Can Siper<br>Funda Durupinar<br>Özgün Babur<br>John A Bachman<br>Benjamin M Gyori<br>Emek Demir |
| Defense Advanced Research Projects Agency | ARO W911NF-15-1-054 | Metin Can Siper<br>Funda Durupinar<br>Özgün Babur<br>John A Bachman<br>Benjamin M Gyori<br>Emek Demir |

The funders had no role in study design, data collection and interpretation, or the decision to submit the work for publication.

**Decision letter and Author response**
Decision letter https://doi.org/10.7554/eLife.68292.sa1
Author response https://doi.org/10.7554/eLife.68292.sa2

# Additional files

## Supplementary files

• Transparent reporting form

## Data availability

All Biofactoid data are available under the Creative Commons CC0 public domain license. To download the data and code, please refer to the documentation on the Biofactoid GitHub repository (github.com/PathwayCommons/factoid). More information on software availability is available in Materials and methods.

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
