## [Decision Letter]

**Decision letter after peer review:**

Thank you for submitting your article "Capturing scientific knowledge in computable form" to *eLife* for consideration as a Feature Article. Your article has been reviewed by three peer reviewers, and the evaluation has been overseen by two members of the *eLife* Features Team (Helena Pérez Valle and Peter Rodgers).

The reviewers and editors have discussed the reviews and we have drafted this decision letter to help you prepare a revised submission.

Summary:

The manuscript of Wong et al., introduces Biofactoid, a novel and intuitive tool to collect interaction data from publications. The manuscript text is clear and provides both a general background for non-specialists as well as it contains the key technical details for experts. The figures nicely illustrate Biofactoid and its application. The authors not only developed the tool but also carried out multiple tests and pilots with users, editors and invited authors. All these phases are properly documented and presented in the manuscript. However, a number of points need to be addressed to make the article suitable for publication.

Essential revisions:

1. Please revise the title so that it more precisely describes the current capabilities of Biofactoid and the use cases demonstrated.

2. The amount of information collected for an interaction (that is, after the drawing just the type of molecular event) is not enough to really utilize the power of this approach in other projects. In particular not having an option to add some kind of biological context and detection methods is a missed opportunity. Please consider introducing an optional box for the biological context, in particular to name the tissue/cell line or the condition (healthy, cancer, etc.). This could come very useful when there are conflicting entries (A activates B; A inhibits B). With the biological context it could be clear that both are correct, otherwise it would look like the same interactors but different signs for the interaction and no further information. As for the type of detection, integrating the Molecular Interactions Controlled Vocabulary (https://www.ebi.ac.uk/ols/ontologies/mi) and facilitating data entry with autocomplete functions would provide useful extra information without overly increasing the time it takes to make a submission. Please consider updating Biofactoid to include this functionality. Please also clarify what ontology (if any) is used for relationships in Biofactoid.

3. Please highlight the fact that submissions to Biofactoid can be edited by users later on using the link sent in the original email. This may be useful if users make a mistake during the submission process or if they need to update the diagram during the publication process.

4. Please address whether there is a process for removing a submission altogether (e.g. if a submission is found to be incorrect or if a paper is being retracted).

5. The two sub-figures of Figure 2B are not easily reproducible. With https://biofactoid.org/ as an entry point, it was not possible to get any networks spanning more than the small ones extracted from a single article as shown in Figure 2A. With pathwaycommons.org as an entry point, it was not possible to find the interactions given in the figure. Concretely, on https://www.pathwaycommons.org/, trying "senp1 sirt3" and clicking on "Interactions", the result was "No interactions to display". Please provide more detailed instructions so the figures shown can be reproduced, or modify Figure 2B to reflect a network that can be obtained using Biofactoid.

6. Please include a discussion of how the facts stated in the paper, regarding the use of Biofactoid, could be qualitatively or quantitatively assessed. Currently, an author working with Biofactoid at all is taken as a successful end point, but surely it will be important to verify that the tool works as described and is, therefore, useful to the community. An assessment could be done, for example, in comparison to professional curators' work, as is often done for text mining approaches.

7. Please include the overall number of recorded facts generated over the multi-step process, including the number of papers used and the facts per paper.

8. Please temper your statements regarding the search and exploration functions of the system. Currently biofactoid.org seems to have no query interface. Since the data is not yet in PathwayCommons, interactive access is effectively only by browsing the papers shown in the landing page, but the figures and text imply much more functionality than is actually available to the user at this point.

9. Please include a comparison to related approaches, considering the citations referenced in line 235 of the manuscript.

10. Please state under what license contributors to Biofactoid are contributing their data, and make it clear both in the article and also in the website, so that users know how they can (re)use the data.

11. Please provide each of the Figures in your manuscript as a separate figure file.

---

## [Author Response]

Essential revisions:1. Please revise the title so that it more precisely describes the current capabilities of Biofactoid and the use cases demonstrated.

Our title has been amended to emphasize the role of authors in pathway curation:

“Author-sourced capture of pathway knowledge in computable form using Biofactoid”

2. The amount of information collected for an interaction (that is, after the drawing just the type of molecular event) is not enough to really utilize the power of this approach in other projects. In particular not having an option to add some kind of biological context and detection methods is a missed opportunity. Please consider introducing an optional box for the biological context, in particular to name the tissue/cell line or the condition (healthy, cancer, etc.). This could come very useful when there are conflicting entries (A activates B; A inhibits B). With the biological context it could be clear that both are correct, otherwise it would look like the same interactors but different signs for the interaction and no further information.

We have updated the Biofactoid curation tool to include a context input box that encourages authors to “*List terms that describe the context (e.g. T cell, cancer, genome stability)*”. Allowing authors to provide a list in free text form has the benefit of accommodating any terms or concepts they consider important for specifying the research context. This information is presented verbatim alongside the network in the Biofactoid Explorer and included in the underlying BioPAX ontology representation, which is also available in Biofactoid’s export files. As a future direction, we will explore ways, such as using natural language processing, to better structure this contextual information without requiring the user to learn the details of complex controlled vocabularies. Users can try the context input box through the Biofactoid curation tool demonstration website at https://biofactoid.org/demo.

As for the type of detection, integrating the Molecular Interactions Controlled Vocabulary (https://www.ebi.ac.uk/ols/ontologies/mi) and facilitating data entry with autocomplete functions would provide useful extra information without overly increasing the time it takes to make a submission. Please consider updating Biofactoid to include this functionality.

Biofactoid aims to capture biological pathway knowledge supported by evidence described in a research article. In this case, authors are free to describe an interaction that represents the findings of one experiment or a set of different experiments. Because entering this information using controlled vocabularies like MI could be complex, even for a trained database curator, we prefer to let authors work with easy to enter plain text data entry in this case (e.g. in the context field). In the future, we will consider ways to help authors annotate the experimental evidence supporting their interactions, for instance by listing the relevant MI terms or by using natural language processing, while still maintaining an easy to use curation interface.

Please also clarify what ontology (if any) is used for relationships in Biofactoid.

Relationships in Biofactoid are mapped to the standard BioPAX ontology that our team developed and that is in use by most major pathway database groups. BioPAX has controlled vocabulary terms for major interaction types (e.g. molecular interaction, catalysis), and extends this to use more specific types by reusing the standard molecular interaction (MI) controlled vocabulary “interaction type” subtree (e.g. “phosphorylation reaction”). We use these in Biofactoid.

3. Please highlight the fact that submissions to Biofactoid can be edited by users later on using the link sent in the original email. This may be useful if users make a mistake during the submission process or if they need to update the diagram during the publication process.

The concluding paragraph of the “Data sharing workflow” section has been updated accordingly:

“Submitted pathways are shared publicly with the research community and users retain the ability to edit their pathway via the link they received at the start of the workflow.”

4. Please address whether there is a process for removing a submission altogether (e.g. if a submission is found to be incorrect or if a paper is being retracted).

The final sentence of the “Data sharing workflow” section has been updated accordingly:

“Authors who wish to have their submissions removed from Biofactoid can do so by contacting the support team (support@biofactoid.org).”

Also, the Biofactoid Explorer web app has been updated so that if PubMed identifies an article as retracted, then users will see a “Retracted Publication” message for that article’s record in Biofactoid.

5. The two sub-figures of Figure 2B are not easily reproducible. With https://biofactoid.org/ as an entry point, it was not possible to get any networks spanning more than the small ones extracted from a single article as shown in Figure 2A. With pathwaycommons.org as an entry point, it was not possible to find the interactions given in the figure. Concretely, on https://www.pathwaycommons.org/, trying "senp1 sirt3" and clicking on "Interactions", the result was "No interactions to display". Please provide more detailed instructions so the figures shown can be reproduced, or modify Figure 2B to reflect a network that can be obtained using Biofactoid.

The results in Figure 2B are intended to demonstrate the utility of information added to Biofactoid. In particular, we show that interactions contributed by two different authors could be used to bridge otherwise disjoint mitochondrial pathways in Pathway Commons.

To create Figure 2B, we manually searched public data in Pathway Commons and Biofactoid with genes of interest and loaded them into the free Cytoscape software for network visualization (cytoscape.org). As a specific example, we executed the following steps to generate the first network in Figure 2B:

1. Visit pathwaycommons.org and search for genes that are part of interactions in Figure 2A: SENP1, SIRT3, UCP1, SIRT5

2. In the search results, listed under “Pathways”, select the Reactome pathways entitled “SUMO is proteolytically processed”, “Transcriptional activation of mitochondrial biogenesis” and “The fatty acid cycling model”. Download a Simple Interaction Format (SIF) file for each. This downloads all of the Reactome pathway information for these genes.

3. Also in the search results, select “Interactions” and download it in SIF. This downloads interactions from many databases involving these genes.

4. Using Cytoscape, import each SIF file as a new network then merge the individual networks using Cytoscape’s merge tool. Manually add the two Biofactoid interactions in Figure 2A to the merged network within Cytoscape.

The description of how these figures were created has been added to the section "Visualization of network data across sources" in Materials and methods.

In the future, we would like to develop additional features for Biofactoid or Cytoscape such that the steps used to create the networks in Figure 2 could be more easily reproduced using automated tools. Specifically, we are working to integrate Biofactoid data continually in Pathway Commons to make querying across all publicly available pathway data easier.

6. Please include a discussion of how the facts stated in the paper, regarding the use of Biofactoid, could be qualitatively or quantitatively assessed. Currently, an author working with Biofactoid at all is taken as a successful end point, but surely it will be important to verify that the tool works as described and is, therefore, useful to the community. An assessment could be done, for example, in comparison to professional curators' work, as is often done for text mining approaches.

In the Discussion (paragraph 6) we have described how Biofactoid currently supports authors in high-quality self-curation, followed by a proposal for efficiently monitoring and improving data quality. Briefly, we will aim to:

Monitor data accuracy using trained curators:

– Recruit at least two curators experienced in collecting pathway knowledge (e.g. Reactome curators)

– Curators independently rate a sample of records with regards to accuracy of entity grounding, interactions, as well as record completeness

– The curator’s ratings are compared to each other to estimate variability

Support community-based data accuracy monitoring:

– Encourage researchers with questions or concerns about a pathway in Biofactoid to communicate directly with the author(s) of the article to resolve issues and, if necessary, the authors can edit the pathway to fix the issues

– Enable researchers to directly report an issue to the Biofactoid support team

– Reserve the right for the Biofactoid support team to lock pathway editing, flag it as disputed or remove it from the collection

7. Please include the overall number of recorded facts generated over the multi-step process, including the number of papers used and the facts per paper.

At the time of writing, Biofactoid contains data for 73 articles in which a total of 330 entities (263 unique) from 7 organisms (*H. sapiens, M. musculus, D. rerio, S. cervisiae, D. melanogaster, C. elegans, E. coli*) are involved in 216 interactions (~3 interactions/article). This information has been included in the last paragraph of the Introduction.

8. Please temper your statements regarding the search and exploration functions of the system. Currently biofactoid.org seems to have no query interface. Since the data is not yet in PathwayCommons, interactive access is effectively only by browsing the papers shown in the landing page, but the figures and text imply much more functionality than is actually available to the user at this point.

We have altered statements from our initial submission that more precisely reflect the existing capabilities of Biofactoid.

– Abstract: “The resulting data is interoperable with public information resources, allowing biological facts from different research reports to be unambiguously connected and making it possible for author-curated knowledge to be appreciated in the context of all existing computable knowledge.”

– Data sharing and exploration: “Beyond the “Explorer”, Biofactoid data represented in the formal BioPAX ontology is interoperable with external pathway and interaction knowledge using technologies for BioPAX processing and analysis, including Paxtools (Demir et al., 2013) and cPath2 (Cerami et al., 2006). This permits author-contributed information to be connected to existing information, and leverage tools for searching, visualizing and analyzing data across different sources.”

– Figure 2

– Legend title: “Biofactoid data is connected to information sources and establishes a bridge between related pathways”.

– Legend: “Biofactoid data establishes a bridge between related pathways described by structured biological knowledge from distinct data sources.”

– Figure 2B, subtitle: “Bridging related pathways.”

9. Please include a comparison to related approaches, considering the citations referenced in line 235 of the manuscript.

A summary and comparison of related approaches is presented in Table 2 which is referenced in the Discussion. This table includes projects that support community curation of biological pathway and interaction knowledge. Given this refined criteria, we have removed references to NDEx (Pratt et al., 2015) and GraphSpace (Bharadwaj et al., 2017), since they are primarily concerned with network information storage and dissemination; removed the reference to Wikidata (Waagmeester et al., 2020) as it only stores metadata for pathways imported from elsewhere (e.g. Reactome, WikiPathways); and removed the reference to INDRA-IPM (Todorov et al., 2019) as it does not involve data storage.

10. Please state under what license contributors to Biofactoid are contributing their data, and make it clear both in the article and also in the website, so that users know how they can (re)use the data.

We have added a "Data availability section to Materials and methods that indicates all contributed data is made freely available without restriction under a Creative Commons CC0 public domain licence. Likewise, the Biofactoid homepage now states that “All Biofactoid data is made freely available to download to the research community under a public domain-equivalent license (Creative Commons CC0 license)” (under section “By researchers, for researchers'').

11. Please provide each of the Figures in your manuscript as a separate figure file.

Each figure is provided as a JPEG file.